# Enhancing Bread’s Benefits: Investigating the Influence of Boosted Native Sourdough on FODMAP Modulation and Antioxidant Potential in Wheat Bread

**DOI:** 10.3390/foods12193552

**Published:** 2023-09-24

**Authors:** Ewa Pejcz, Sabina Lachowicz-Wiśniewska, Paulina Nowicka, Agata Wojciechowicz-Budzisz, Joanna Harasym

**Affiliations:** 1Department of Biotechnology and Food Analysis, Wroclaw University of Economics and Business, 53-345 Wroclaw, Poland; agata.wojciechowicz-budzisz@ue.wroc.pl (A.W.-B.); joanna.harasym@ue.wroc.pl (J.H.); 2Department of Health Sciences, Calisia University, 4 Nowy Świat Street, 62-800 Kalisz, Poland; sabinalachowicz90@gmail.com; 3Department of Fruit, Vegetable and Nutraceutical Technology, Wrocław University of Environmental and Life Sciences, 51-630 Wrocław, Poland; paulina.nowicka@upwr.edu.pl

**Keywords:** sourdough fermentation, wheat bread, antioxidant activity, FODMAP

## Abstract

This study aimed to assess the impact of bacterial species and fermentation time on wheat bread quality, FODMAP (fermentable oligosaccharides, disaccharides, monosaccharides, and polyols) content, and antioxidant activity of wheat bread, utilizing boosted native sourdough as a novel approach to enhance bread production. The incorporation of lactic acid bacteria strains, i.e., *Lacticaseibacillus casei* and *Lactiplantibacillus plantarum*, during 72 h fermentation significantly reduced FODMAP content to less than 0.1 g/100 g of wheat bread. Extending the fermentation time to 72 h notably increased the polyphenol content to 145.35 mg gallic acid (GA) per 100 g in the case of spontaneous fermentation and to 151.11 and 198.73 mg GA/100 g in the case of sourdoughs inoculated with *L. casei* and *L. plantarum*, respectively. While the treatment yielded positive effects on FODMAP modulation and antioxidant activity, it is crucial to acknowledge its impact on some organoleptic properties, such as aroma and flavor, which, despite good overall bread quality, have changed as a result of prolonged fermentation time. The study results indicate that choosing specific bacterial species and controlling fermentation time can effectively reduce FODMAPs and boost antioxidants. These findings contribute to the understanding of sourdough-based interventions in bread production, offering insights for the development of healthier and nutritionally improved wheat bread products.

## 1. Introduction

Carbohydrates representing simple sugars and certain oligosaccharides, i.e., sugars consisting of 3–10 sugar moieties, are referred to as FODMAPs (Fermentable Oligosaccharides, Disaccharides, Monosaccharides, and Polyols). While these carbohydrates can have positive effects on the human body, their incomplete digestion may lead to various gastrointestinal symptoms [1]. Certain amounts of compounds belonging to this group may elicit a positive effect on the digestive tract, while their excess may cause various abdominal ailments and symptoms, such as diarrhea, flatulence, constipation, or abdominal pain [1,2,3,4]. Excessive consumption of food products with a high FODMAP content (over 20 g/day) may lead to the accumulation of sugars in the intestines. The related symptoms are mild in healthy persons but much worse tolerated by patients suffering from irritable bowel syndrome (IBS) [4]. IBS is a common gastrointestinal disorder that can impact individuals of all age groups, genders, and ethnic backgrounds, with a prevalence ranging from 5% to 20%. It is often observed to disproportionately affect young individuals, particularly women [2,3].

To alleviate IBS symptoms, a low-FODMAP diet is often recommended, which involves restricting the intake of short-chain carbohydrates [3]. Wheat bread, known for its high fructan content, is typically eliminated from the diet in favor of gluten-free alternatives [2,5]. Nevertheless, the utilization of sourdough in the bread-making process offers a promising avenue to diminish FODMAP levels [6,7,8]. Given the metabolic characteristics of sourdough bacteria and their secondary metabolites, this approach aims to modify the enzyme profile through targeted engineering. Consequently, it not only responds to the need for effective strategies aimed at reducing FODMAP content in bread but also improves our understanding of their metabolic potential, creating new possibilities for innovative food product design.

Sourdough, an ecosystem of yeast cells and lactic acid bacteria, not only enhances the nutritional value, taste, aroma, texture, and storage stability of bread but also induces changes in carbohydrate composition [9]. The selection of lactic acid bacteria strains, such as *Lactobacillus delbrueckii*, *Lacticaseibacillus casei*, *Lactiplantibacillus plantarum*, and *Ligilactobacillus salivarius*, further enhances the metabolic activity of the fermentation microflora [10,11]. *L. casei* and *L. plantarum* are preferred strains for sourdough bread production due to their ability to create the tangy flavor, enhance shelf life through acidity, improve digestibility, and release nutrients from grains. They also act as natural preservatives, contribute to a desirable texture, and offer customizable flavor profiles. Their historical use in sourdough bread aligns with the tradition of natural fermentation methods. Sourdough-based bakery products exhibit improved sensory qualities, digestibility, availability of micro- and macro-elements, preservation of bioactive compounds, and enhanced antioxidant properties [12,13]. The antioxidant activity of sourdough components depends on the specific inoculum used for fermentation and the duration of sourdough fermentation; it can be further enhanced by promoting the formation of potentially antioxidative peptides through the metabolic activity of LAB during acidic fermentation, resulting in increased phenol content and antioxidative activity via acidification and hydrolysis of more complex and glycosylated forms [14,15].

The use of sourdough in bread production offers potential benefits; however, there is limited knowledge regarding the specific effects of bacterial species and fermentation time on these parameters. The primary objective of this study was, therefore, to investigate the impact of two different bacterial species used for sourdough fermentation and the duration of fermentation on the quality of wheat bread, FODMAP content, and antioxidant activity. In addition, it aimed to explore the potential of utilizing boosted native sourdough, which allows for targeted reductions in FODMAP content and provides insights for the development of healthier wheat bread products with enhanced antioxidant activity. The novelty of this research lies in its comprehensive approach, utilizing wheat sourdough bread and a one-strain inoculum to investigate specific metabolic activity, enabling tracked changes in flour components with the aim of optimizing the fermentation process to enhance the nutritional value and consumer acceptability of wheat bread.

## 2. Materials and Methods

### 2.1. Materials

The research material included type 650 wheat flour from GoodMills Sp. z o.o. (Stradunia, Poland) and two strains of lactic acid bacteria from the German Collection of Microbial Strains and Cell Cultures (Deutsche Stammsammlung für Mikroorganismen und Zellkulturen; Braunschweig, Germany): *Lacticaseibacillus casei* (DSM 20011) and *Lactiplantibacillus plantarum* subsp. *plantarum* (DSM 20174). Commercial pressed yeast *Saccharomyces cerevisiae*, salt, and tap water were also used for bread baking. 

### 2.2. Flour Quality

The flour quality was assessed using the AACC methods [16] based on the fraction size, the falling number—using the Hagberg–Perten method (56-81.04 method), and the protein content—using the Kjeldahl method (46-12.01 method). The flour was also assessed for its rheological parameters using a Mixolab apparatus (Chopin Technologies, Villeneuve-la-Garenne, France) (Method 54-60.01) and the Chopin+ protocol, determining the torque at characteristic points. All flour parameters are presented as the mean values from two replications.

### 2.3. Sample Preparation

The sample preparation process consisted of the multiplication of microorganisms, preparation and fermentation of sour soups, production and fermentation of sourdough, and bread baking. The bacteria of the species *L. casei* and *L. plantarum* were multiplied in a liquid culture using the MRS medium, which has components selective for the growth of lactic acid bacteria, at the temperature of 40 °C. Then, wheat flour was used to prepare appropriate sour soups, consisting of flour (100%), water (300%), and liquid culture of microorganisms (10^9^ CFU/kg of flour), that were fermented for 72 h at the temperature of 28 °C. Spontaneously fermenting sour soups were made without the microbial culture addition. The obtained sour soups were used to prepare sourdoughs consisting of appropriate flour (100%), water (100%) and sour soup (10%), which were fermented for 24, 48, and 72 h at 30 °C. In the next stage, three doughs were prepared for each variant using the sourdoughs produced. For the tested samples, the first phase was sourdough (constituting 50% of the total flour included in the recipe); in the second phase, the sourdough was combined with the remaining flour (50%) and water, and with yeast (3%) and salt (1.5%). The control samples consisted of wheat bread prepared using the single-phase method (without sourdough). Dough was prepared in a farinograph (Brabender, Duisburg, Germany); then, pieces of each dough variant were fermented for 60 min at 30 °C in a fermentation chamber (IBIS, Szubin, Poland). The bread was baked for 30 min at 220 °C in an IBIS GT 600 oven (IBIS, Szubin, Poland).

### 2.4. Bread Quality

Fresh bread was assessed for volume and crumb color, and subjected to consumer acceptance assessment. The bread volume was established using the SA-WY apparatus (ZBPP, Bydgoszcz, Poland) by measuring the difference in the amount of millet before and after inserting a loaf of bread into the charging chamber of the device. The bread loaf volume is presented as the average of three measurements for each loaf per 100 g of flour. The color of the crumb was determined using a Minolta Chroma Meter CR-200b colorimeter (Konica Minolta, Tokyo, Japan), based on the CIE Lab color model including three parameters: *L**, *a**, and *b**. The lightness parameter *L** = 0 means black, while *L** = 100 means white. Both of the *a** and *b** parameters have two values: positive and negative, denoting appropriate colors. The green color is indicated by the *a**− parameter, while the red color is indicated by the *a**+ parameter. The *b**− parameter denotes yellow, while *b**+ means blue. Crust and crumb color measurement results are presented as the mean of five replicates. In addition, the color parameters delta E (ΔE), hue (H), chroma (C*), browning index (BI), and delta browning index (ΔBI) were calculated according to the following formulas: (1)$$\Delta E=\sqrt{{\left(L^{*}-L_{0}^{*}\right)}^{2}+{\left(a^{*}-a_{0}^{*}\right)}^{2}+{\left(b^{*}-b_{0}^{*}\right)}^{2}}$$
where *L*_0_, *a*_0_, and *b*_0_ are color parameters of control bread,
(2)$$H=\arctan(\frac{b^{*}}{a^{*}}),$$



(3)
$$C*=\sqrt{{\left(a^{*}\right)}^{2}+{\left(b^{*}\right)}^{2}},$$


(4)
$$\mathrm{BI}=\;\frac{100\left(x-0.31\right)}{0.17},\;\mathrm{where}\;x=\frac{a^{*}+1.75L^{*}}{5.645L^{*}+a^{*}-0.3012b^{*}},$$



ΔBI = BI − BI_0_
(5)
 where BI_0_ is browning index of control bread.

The tested wheat breads were subject to consumer acceptance assessment carried out using a 9-point hedonic scale. The evaluators scored 9 points for the most desirable quality attributes and 1 point for the least desirable ones. The assessment included the following bread quality attributes: external appearance, crumb porosity, crumb consistency, aroma, and taste. The organoleptic panel consisted of 14 men and women, aged 22–58. Apart from being subjected to quality analysis, the cooled bread was prepared for further analyses, i.e., freeze-dried, ground in a WŻ-1 mill (ZBPP, Bydgoszcz, Poland), and vacuum-packed. 

### 2.5. FODMAP Content

The content of FODMAP compounds and total polyphenol content (TPC) as well as antioxidant activity were determined using the samples of flour and bread. The content of fructans was determined acc. to the AACC 32-32.01 spectrophotometric–enzymatic method [16], which is based on the determination of fructose from enzymatically-degraded fructans by reading the absorbance at 410 nm. This determination was performed in duplicate. The content of fermentable sugars was determined using high-performance liquid chromatography (HPLC). The samples (10–15 g) were mixed with distilled water and heated for 20 min at the temperature of 100 °C. After cooling, they were centrifuged at 20,000× *g* revolutions for 10 min. In order to purify the samples, i.e., from proteins, they were filtered using Sep-Pak filters and washed with distilled water. The purified sample (40 µL) was injected using an autosampler (L-7200) into a 3 µL Unison UK-Amino column (Imtakt, Kyoto, Japan). The determination was performed using a PL-ELS 1000 Evaporative Light Scattering Detector with the following input parameters: evaporator temperature—80 °C, nebulizer temperature—80 °C, and nitrogen flow—1.2 SLM. The sugars were eluted at 30 °C in an isocratic flow using an 85% acetonitrile solution at a flow rate of 0.7 mL/min. The determined sugars, including glucose, fructose, lactose, kestose, mannitol, and sorbitol, were identified using respective standards. From among the determined sugars and polyols, only fructose and mannitol were identified in the bread samples. The content of FODMAPs is presented as the sum of fructans and identified sugars and polyols, and expressed in grams/100 g of the product.

### 2.6. Antioxidant Properties

For the antioxidant activity determination, the samples were prepared according to the method described by Lachowicz et al. (2021) [17]. The total polyphenol content of the samples was determined using the Folin–Ciocalteu spectrophotometric method [18] with absorbance measured at 765 nm after 1 h using a UV-2401 PC spectrophotometer (Shimadzu, Kyoto, Japan). Results are expressed as mg gallic acid equivalents (GAE) per 100 g dry sourdough. Data are expressed as the mean of three measurements. The anti-radical activity measured using ABTS^•+^assay and the FRAP assay, which measures the antioxidant activity of compounds capable of reducing the ferric complex, were conducted following the protocols outlined by Re et al. (1999) [19] and Benzie and Strain (1996) [20]. Absorbance was measured at 734 nm and 593 nm using a UV-2401 PC spectrophotometer (Shimadzu, Kyoto, Japan). The results are expressed as Trolox equivalents in mmol/L per 100 g dry sample. Data are expressed as the mean of three measurements.

### 2.7. Statistical Analysis

The results were analyzed statistically using the Statistica 13.3 software package (StatSoft, Tulsa, OK, USA). Standard deviation was calculated for flour quality parameters. The statistical analysis utilized a two-way analysis of variance (ANOVA) at *p* = 0.95 with fermentation time and sourdough type as factors. Homogeneous and significantly different groups were determined using Duncan’s post hoc tests.

## 3. Results and Discussion

### 3.1. Flour Quality

Type 650 wheat flour was characterized by fine fraction size (93 μm), low α-amylase activity measured using the falling number (390.5 s), and appropriate protein content (14.72%) (Table 1). The rheological properties of type 650 wheat flour were determined using the Mixolab apparatus which enables examining the properties of the protein-starch complex at the varying dough mixing temperature. The water absorption of wheat flour corresponding to a dough consistency of 1.1 Nm was 59.10%. The time needed to reach the C1 point of the diagram characterizing the dough development time was short and amounted to 3.38 min, and the dough stability time was 8.40 min. The reduction in dough resistance at point C2 was caused by the loosening of its structure due to its mixing and temperature increase. The dough resistance at point C3 characterizes the maximum viscosity of the dough as an effect of starch pasting, while point C4 characterizes the amylolysis phase. The retrogradation phase was characterized by the data achieved in point C5. The results obtained show that the tested flour was strong and characterized by high stability of starch paste and low activity of amylolytic enzymes [21].

### 3.2. Bread Quality

The qualitative parameters of wheat bread depending on the time and type of fermentation are presented in Table 2. The lowest loaf volume per 100 g of flour was found for the breads with the addition of spontaneously fermenting sourdough and for the bread made without sourdough. The breads made with sourdough inoculated with both *L. casei* and *L. plantarum* had a higher loaf volume (on average 502.6 and 507.3 cm^3^ per 100 g of flour for *L. casei* and *L. plantarum*, respectively) in comparison to spontaneously fermented sourdough bread (average 450.0 cm^3^); however, their loaf volume decreased along with the extension of the fermentation time of these sourdoughs, reaching the highest value after 24 h of fermentation. The wheat bread volume depends both on the time of sourdough fermentation and the fermentation method. According to Crowley et al. (2002) [22], wheat bread with sourdough added had a similar or higher volume compared to the bread produced without sourdough. In turn, Struyf et al. (2017) [23] demonstrated that breads made from spontaneously fermented sourdough with *S. cerevisiae* yeast had an optimal or reduced bread volume, related to the ability of microorganisms to produce carbon dioxide, which loosens dough structure and increases its volume. Sourdoughs that were fermented using lactic acid bacteria of the *L. casei* and *L. plantarum* species contributed to the higher loaf volume of the breads made with their addition. These results are consistent with findings reported by Dal Bello et al. (2007) [24] and Menezes et al. (2018) [1], who claimed that sourdough addition to bread increased the volume of baked loaves. The sourdough fermentation time of 24 h turned out to be optimal to obtain bread with an increased loaf volume. The *L** and *a** parameters of the crumb color also differed depending on the type and time of sourdough fermentation. The longer the sourdough fermentation time, the darker the bread was (lower *L** values). Regardless of the sourdough fermentation method, the darkest color of the crumb was found in the breads made from the sourdough fermented for 72 h, while the lightest color was found in the breads made with the addition of the control bread without sourdough. Taking into account the method of fermentation, the breads made from sourdough fermented with *L. plantarum* had the darkest color, while the crumb of spontaneously fermented sourdough bread was the lightest. The *a** parameter, depending on the value it takes, denotes either red or green color. The results obtained indicate that the chromaticity of the green color of the bread crumb decreased with extending sourdough fermentation time. The chromaticity of green color of the bread crumb was higher for spontaneously fermented sourdough bread than when inoculum was used. The *b** parameter responsible for the contribution of yellow and blue reached lower values in the case of sourdough bread than in the control bread. Higher values of this parameter (more yellow hue) were also observed in the case of using spontaneous fermentation than of the inoculated fermentation. The change in the color of the crumb presented as ΔE was greater the longer the sourdough fermentation time was. Higher values were obtained with inoculated fermentation (especially *L. plantarum*) than with spontaneous fermentation. The extended fermentation time of the sourdough as well as its inoculation resulted in lower hue and chroma values and higher browning index values. The change in the browning index compared to the control bread (Figure 1) indicates that the greatest change in this parameter is due to the first 24 h of fermentation, most with *L. plantarum* inoculum and least with spontaneous fermentation. The longer the fermentation time, the smaller the differences between the bread with the addition of different types of sourdough.

The baked wheat breads were subjected to a consumer acceptance assessment carried out using a nine-point hedonic scale. The panelists assessed various quality attributes of the bread: external appearance (Figure 2), crumb porosity, texture, aroma and taste (Figure 3). With the longer fermentation time of the sourdough, only crumb texture was rated higher, while flavor ratings deteriorated. The quality parameters of bread with spontaneously fermented sourdough and inoculated with *L. casei* were assessed similarly, while the use of *L. plantarum* inoculum resulted in lower scores for porosity, aroma, and flavor of the bread. Gänzle (2014) [25] believes that the fermentation carried out by lactic bacteria for 8 h to 144 h produces an appropriate amount of endogenous enzymes, enabling changes to be made at the stage of bread dough making, which in the next stage will allow us to obtain bread with the best consistency. On the other hand, Un-Nisa et al. (2016) [26] claim that the products of sourdough fermentation are responsible for the pleasant aroma of wheat bread, and the longer the sourdough fermented time is, the more pleasant is the aroma of the bread produced from sourdough fermented by *L. casei*. According to Dal Bello et al. (2007) [24], metabolites of *L. plantarum* fermentation are responsible for improving the aroma of wheat bread. In the case of sourdough fermented for 24 h by lactic acid bacteria, the aroma of wheat bread made from sourdough fermented by *L. casei* was described as cheese-like, while the pungent aroma was characteristic of the bread made from sourdough fermented by *L. plantarum*. The perception of such aromas by panelists may be due to the production of metabolites during LAB fermentation. According to the participants of the organoleptic panel, the breads made from sourdough fermented by *L. plantarum* had a sour taste, the intensity of which increased with the extending sourdough fermentation time. Un-Nisa et al. (2016) [26] indicate that mainly lactic acid bacteria are responsible for the sour taste of bread.

### 3.3. FODMAP Content

Fructans constituted the dominant FODMAP fraction in each type of bread and no glucose was identified in any of the bread samples (Figure 4). Fructose turned out to be the major sugar found in the produced breads. Its highest content was found in the breads made of the sourdough fermenting spontaneously for 24 h (0.0195 g/100 g). It was also detected in the bread produced without sourdough (0.0126 g/100 g), in bread made of the sourdough fermenting spontaneously for 48 h (0.0068 g/100 g), and in that made of sourdough fermented with *L. casei* for 24 h (0.0056 g/100 g). Trace amounts of mannitol were identified only in the bread with the addition of *L. casei*-fermented sourdough for 48 h and 72 h. The content of fructans (0.22 g/100 g) and the content of FODMAPs (0.23 g/100 g) were the highest in the bread made without the sourdough. Extending the LAB-assisted fermentation time of the sourdough to 72 h resulted in fructan content reduction to 0.09 g/100 g for the breads made based on the sourdough inoculated with both *L. casei* and *L. plantarum*. In the wheat breads made of the sourdough fermented for 24 h with *L. casei* (0.16 g/100 g) and *L. plantarum* (0.17 g/100 g), the content of fructans was higher than in the bread made from spontaneously fermenting sourdough. In the case of 48 h sourdough fermentation, the highest content of fructans (0.16 g/100 g) was determined in the bread made of the sourdough fermented by *L. casei* bacteria. In the bread made of spontaneously fermenting sourdough and the bread produced from the sourdough fermented by *L. plantarum*, the fructan content was almost the same and reached 0.13 g/100 g and 0.14 g/100, respectively. The fermentation time of 72 h meant that, regardless of the fermentation method, the fructan content was at the same low level, amounting to 0.10 g/100 g in the case of spontaneous fermentation and 0.09 g/100 g in the case of fermentation by *L. casei* and *L. plantarum*. In the case of fermentation assisted by *L. plantarum*, the successive extension of the sourdough fermentation time resulted in a successive reduction of the FODMAPs content. In the case of fermentation carried out by *L. casei*, the highest content of FODMAPs was determined in the bread made of sourdough fermented for 48 h. Regardless of the sourdough fermentation method, the lowest FODMAPs content was found in the breads made of the sourdough fermented for 72 h.

Fructans turned out to be the major FODMAPs found in the analyzed wheat flour, with their content reaching 1.15 g/100 g flour. The FODMAPs content in the tested flour was at the same level. According to Struyf et al. (2017) [23] and Fraberger et al. (2018) [10], the content of fructans in wheat flour is at 1.4–1.7 g/100 g flour. In the present study, it was influenced by both the time and method of sourdough fermentation. Breads made without the sourdough had the highest content of fructans, while the addition of sourdough and the successive extension of its fermentation time to 72 h resulted in fructan content reduction. Loponen and Gänzle (2018) [11], who achieved a fructan content of 0.06 g/100 g in bread made of the spontaneously fermenting sourdough, emphasize that lowering the fructan content is possible owing to the appropriate selection of the sourdough fermentation time and microorganisms, consisting in the appropriate combination of yeast and lactic acid bacteria. Sugars, such as sucrose, maltose, fructose, and glucose, are the first to ferment. Extending the fermentation time allows fructans to be broken down into simple sugars. Owing to this, these sugars can be fermented by microorganisms in the sourdough [11]. Also, in the study by Menezes et al. (2018) [1], the content of fructans in wheat bread decreased along with extending sourdough fermentation time. Apart from fructans, which are the main representatives of FODMAPs, other sugars belonging to this group of compounds are also found in wheat bread. The spontaneous sourdough fermentation resulted in the greatest amount of carbohydrates broken down into fructose, while during the fermentation process conducted by *L. casei*, a facultative heterofermentative LAB, the presence of mannitol was observed due to its fructose metabolism. The presence of mannitol in wheat bread was explained by Ziegler et al. (2016) [8] and Loponen and Gänzle (2018) [11] who believe that it occurs due to the lactic acid bacteria, which are responsible for fructose conversion to mannitol. The lack of glucose in the tested wheat breads may be due to the fact that this sugar is first fermented by lactic acid bacteria. When glucose is depleted, these bacteria begin to ferment residual sugars, such as maltose or fructose [9]. Also, according to Grausgruber et al. (2020) [27], the FODMAPs content reduction in wheat bread can be achieved not only by making bread from the low-FODMAPs flour but also by choosing appropriate bread making method and the most effective method for reducing the FODMAPs level in bread is to use sourdough. The research carried out by Ziegler et al. (2016) [8] shows that, by applying fermentation time of more than 4 h, it is possible to reduce the FODMAPs content in wheat bread by 90% compared to their content in the flour. The present study results show that the FODMAPs content in wheat bread made of the spontaneously fermenting sourdough decreased by 57.8% compared to their content in the control bread and by 91.5% compared to their content in flour, namely by 60.4% and 92.0%, respectively, with the use of *L. casei*-inoculated sourdough and by 62.2% and 92.4% with the use of *L. planatrum*-inoculated sourdough.

### 3.4. Antioxidant Properties

The determined total polyphenol content (TPC) was higher in the bread produced after each type of sourdough fermentation than in the bread produced without sourdough and was higher the longer the fermentation time (Table 3). Sourdough fermentation for 72 h resulted in a polyphenol content increase to 145.35 mg gallic acid (GA) per 100 g in the case of spontaneous fermentation and to 151.11 and 198.73 mg GA/100 g in the case of sourdoughs inoculated with *L. casei* and *L. plantarum*, respectively. According to Banu et al. (2010) [14] the fermentation process may increase the TPC of bread by increasing the amount of easily extractable phenolic compounds. The lowest content of polyphenols was determined in the control bread made without the sourdough (62.32 mg GA/100 g). The use of *L. plantarum*-inoculated sourdough had the greatest impact on increasing the polyphenol content in the breads. An increase in the content of polyphenols with the time of fermentation assisted by *L. plantarum* was also observed by Chiș et al. (2018) [28]. The proteolytic activity of LAB influences the profile of polyphenols, which contributes to the improvement in their solubility [28,29].

Changes in the antioxidant activity of the breads measured using the ABTS^•+^ method showed no significant effect of fermentation time on this parameter but indicated that inoculated sourdoughs contributed to an increase in the antioxidant activity of bread compared to spontaneous fermentation. The FRAP method results were similar to those observed for the total polyphenol content. The activity increased with the extension of sourdough fermentation time; however, this increase was significant within the first 24 h of fermentation, with the increase being greater upon the use of inoculated than spontaneously fermenting sourdoughs. Acid fermentation may lead to the formation of potentially antioxidant bioactive peptides, and fermentation time extension may enhance the antioxidant activity of the products [15]. The bread made of the sourdough inoculated with both *L. casei* and *L. plantarum* had a similar antioxidant activity after 24 h of fermentation to that of bread with sourdough fermenting spontaneously for 72 h. Inoculation of sourdough with *Lactobacillus rhamnosus* bacteria was also reported to enhance the antioxidant activity compared to its spontaneous fermentation (Banu et al. 2010) [14]. The metabolic activity of LAB affects the content of bioactive components, which allows for an increase in the antioxidant activity through the release of antioxidant peptides, which in turn increases the content of phenols and enhances antioxidant activity through acidification and hydrolysis of more complex and glycosylated forms [14,30]. The weak correlation between results of TPC determination and ABTS and FRAP assays may be due to the formation of protein–polyphenol complexes causing falsely low values of antioxidant activity [18].

## 4. Conclusions

Previous research [31] has shown that 72 h fermentation time of *L. plantarum*-inoculated sourdough reduced the FODMAP content by 91% in wheat sourdough and that sourdough fermentation time of at least 72 h also positively influenced the content of polyphenols and antioxidant activity in wheat sourdough, regardless of fermentation type.

Producing wheat bread with a reduced content of FODMAP compounds is feasible by making the dough with the use of sourdough inoculated with lactic acid bacteria of the *L. casei* and *L. plantarum* species for 72 h. Extension of the sourdough fermentation time contributed to FODMAP content reduction, especially fructans. In addition, extension of the lactic acid fermentation contributed to the improvement of the antioxidant properties of wheat bread, which were the highest upon the use of the *L. plantarum*-inoculated sourdough. In turn, as lactic acid fermentation was extended, some organoleptic characteristics of wheat bread deteriorated, including the development of a sour taste. This research presents a novel approach by using wheat sourdough bread and a single-strain inoculum. This helps us track changes in the flour’s components, aiming to make wheat bread healthier and more appealing to consumers. More research is needed for further optimization of the fermentation and baking process to achieve better bread taste and to gain a deeper understanding of the exact changes that occur during fermentation and baking.

## Figures and Tables

**Figure 1 foods-12-03552-f001:**
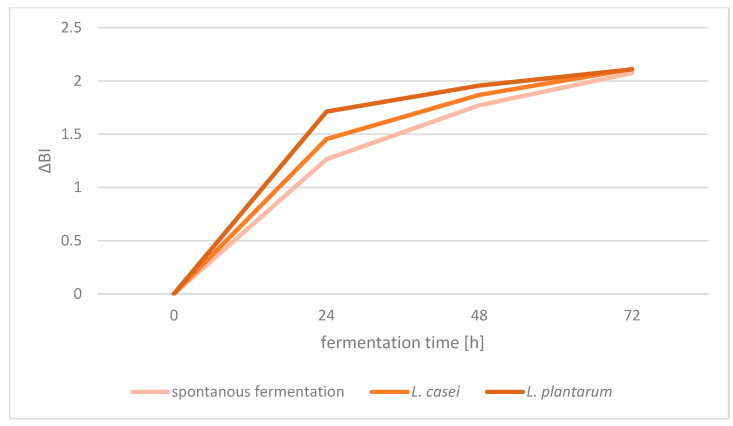
ΔBI of breads depending on the fermentation time.

**Figure 2 foods-12-03552-f002:**
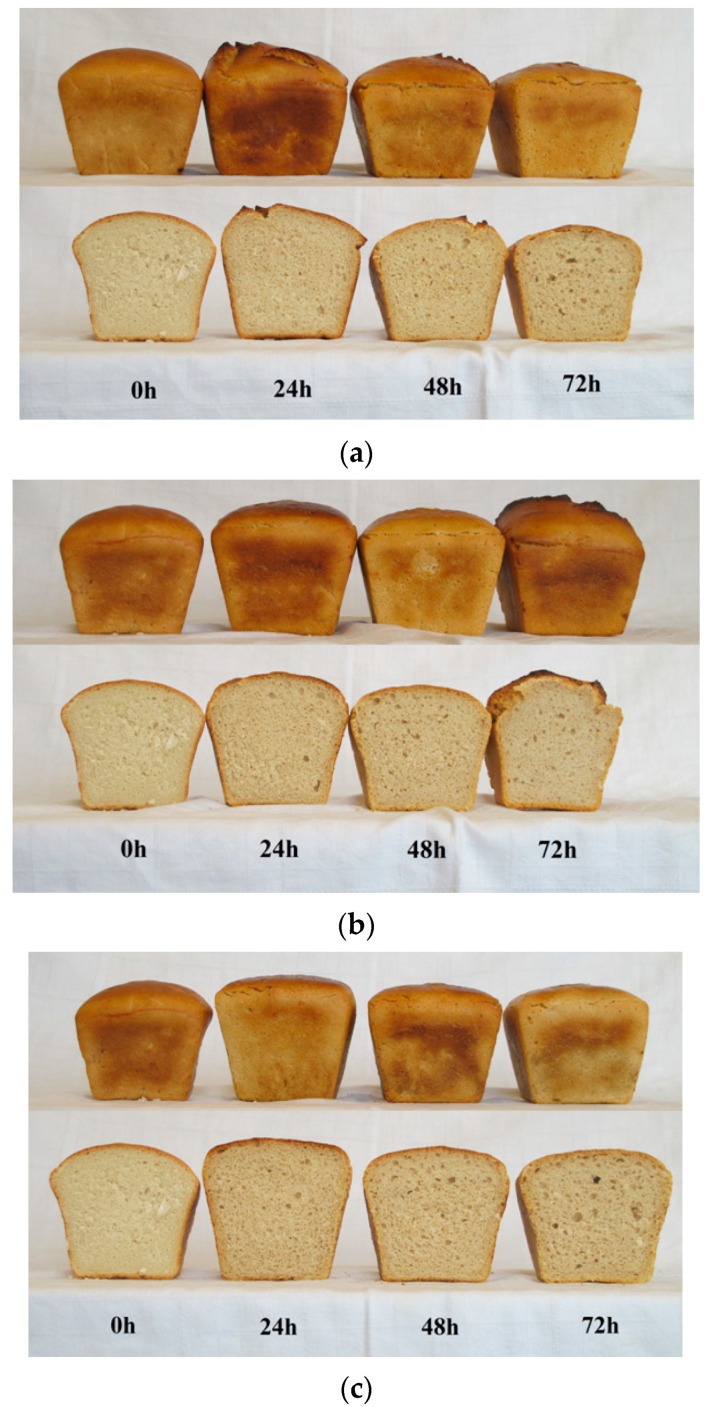
The external and internal appearance of breads depending on the fermentation time: (**a**) Spontaneously fermented; (**b**) *L. casei* inoculated; (**c**) *L. plantarum* inoculated.

**Figure 3 foods-12-03552-f003:**
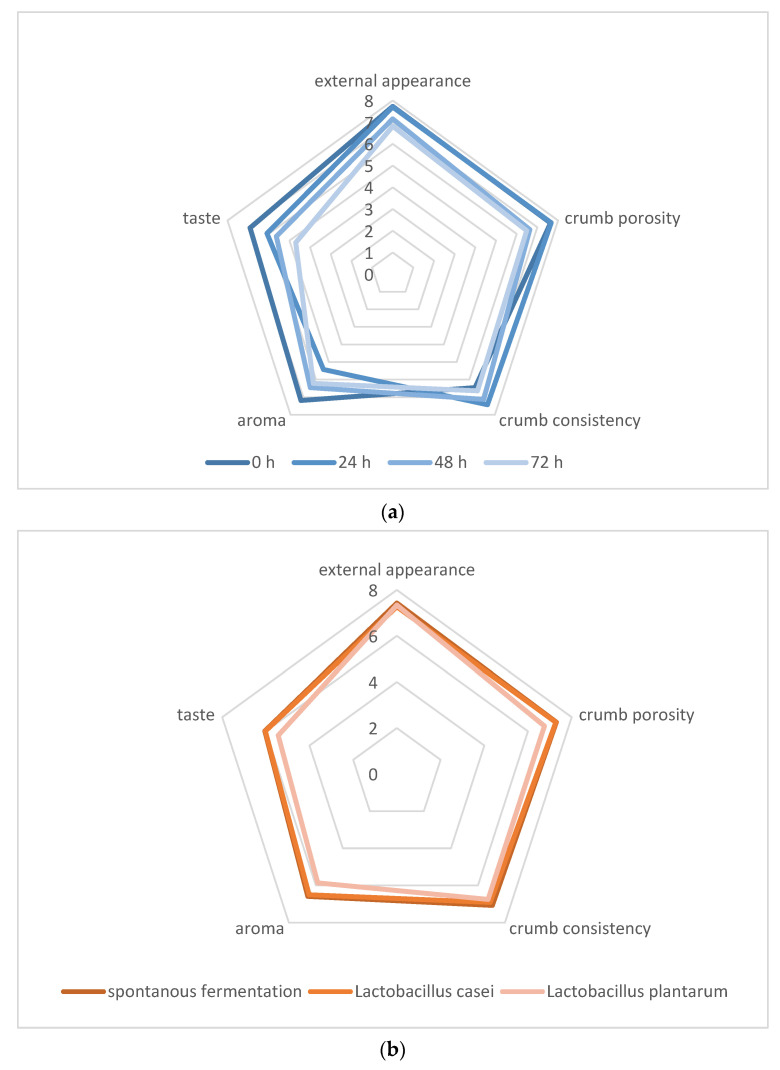
The organoleptic properties of breads depending on fermentation time (**a**) and fermentation type (**b**).

**Figure 4 foods-12-03552-f004:**
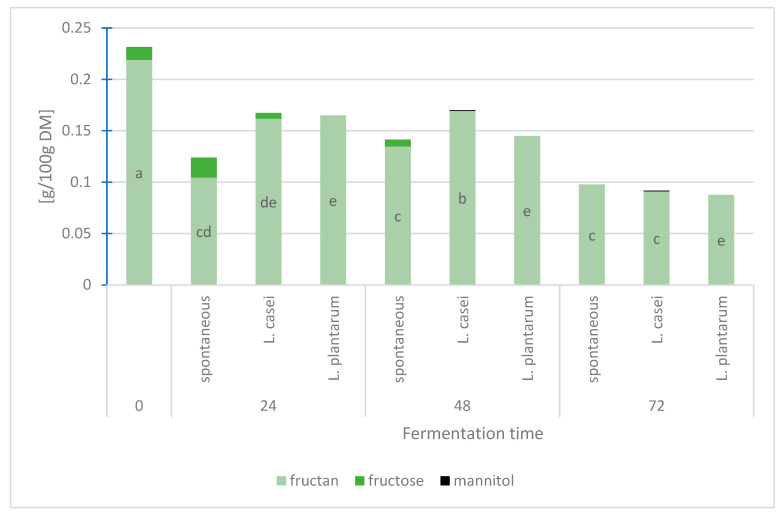
The FODMAPs content in wheat bread [g/100 g DM]. Small letters denote significant differences according to Duncan’s test (*p* ≤ 0.05).

**Table 1 foods-12-03552-t001:** The quality parameters of type 650 wheat flour and the rheological characteristics of its dough.

Fraction Size [μm]	Falling Number [s]	Protein Content [%]	Water Absorption [%]	Dough Development Time [min]	Dough Stability [min]	Dough Softening (C2) [Nm]	Peak Visosity (C3) [Nm]	Activity of Amylolytic Enzymes (C4) [Nm]	Retrogradation (C5) [Nm]
93 ± 0.2	390.5 ± 4.6	14.72 ± 0.20	59.10 ± 0.57	3.38 ± 0.177	8.40 ± 0.141	0.424 ± 0.001	1.656 ± 0.003	1.770 ± 0.034	2.892 ± 0.042

Values represent the means of two replicates ± standard deviation.

**Table 2 foods-12-03552-t002:** The qualitative parameters of wheat bread depending on fermentation time and sourdough type.

Factors		Loaf Volume [cm^3^/100 g]	*L**	*a**	*b**	ΔE	Hue	Chroma	BI
Fermentation time	0 h	456.2 d	72.33 a	−2.98 d	22.06 a	0.00 d	277.66 a	22.25 a	0.01 d
	24 h	505.3 a	66.60 b	−1.41 c	20.46 c	43.21 c	273.90 b	20.50 c	1.50 c
	48 h	474.0 c	65.79 c	−1.16 b	21.12 b	53.54 b	273.10 c	21.15 b	1.89 b
	72 h	480.6 b	64.75 d	−0.89 a	20.49 c	70.57 a	272.45 d	20.50 c	2.13 a
Sourdough type	Spontaneous fermentation	450.0 b	67.84 a	−1.74 b	21.54 a	34.93 c	274.58 a	21.62 a	1.30 c
	*L. casei*	502.6 a	67.37 b	−1.56 a	20.68 c	42.01 b	274.19 b	20.75 c	1.38 b
	*L. plantarum*	507.3 a	66.89 c	−1.52 a	20.86 b	48.80 a	274.06 b	20.93 b	1.47 a

Values represent the means of 3 (loaf volume) or 5 (colour) replicates. Small letters in the same column denote significant differences according to Duncan’s test (*p* ≤ 0.05).

**Table 3 foods-12-03552-t003:** The total polyphenol content (TPC) and the antioxidant activity of the wheat sourdough breads.

Factors		TPC [mgGA/100 g DM.]	ABTS [mmol Trolox/100 g DM.]	FRAP [mmol Trolox/100 g DM
Fermentation time	0 h	62.32 c	0.44 a	0.39 b
	24 h	131.76 b	0.38 b	0.51 a
	48 h	130.51 b	0.44 a	0.53 a
	72 h	157.28 a	0.40 b	0.50 a
Sourdough type	spontaneous	122.31 ab	0.38 b	0.45 b
	*L. casei*	114.35 b	0.44 a	0.51 a
	*L. plantarum*	124.74 a	0.43 a	0.50 ab

Values represent the means of 3 replicates. Small letters in the same column denote significant differences according to Duncan’s test (*p* ≤ 0.05). DM—sample dry matter.

## Data Availability

The datasets used and/or analyzed during the current study are available from the corresponding author on reasonable request.

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
