# Peer review of "Enhancing Bread’s Benefits: Investigating the Influence of Boosted Native Sourdough on FODMAP Modulation and Antioxidant Potential in Wheat Bread"

_foods, 2023, doi:10.3390/foods12193552_

Round 1
Reviewer 1 Report
-
The research presents an interesting topic within the field of food science; however, the manuscript should be adjusted with the following considerations in mind:
Enhance the introduction. A short paragraph should be included to complement the information justifying the use of lactic acid bacteria strains, especially Lacticaseibacillus casei and L. plantarum, and how they can influence the wheat flour fermentation processes, conferring the properties and characteristics outlined in the study. Additionally, it is essential to include the chemical changes that enable the development of compounds with potential antioxidant properties in the fermented dough.
Organize the presentation of the materials and methods section into subsections based on each procedure and activity, rather than presenting it as a single body of text. This organization will improve readability and maintain order. The same observation applies to the presentation of results, which should also be better organized, following the structure outlined in the materials and methods section.
Figure 4 lacks units corresponding to the Y-axis. Furthermore, the discussion regarding the various factors affecting antioxidant activity should be improved. The phenolic content appears to have a low correlation with the presented antioxidant activity, suggesting that it may depend more on other compounds. In this regard, the authors are advised to thoroughly review the explanations provided, aiming for better alignment with the results of the ABTS and FRAP assays.
It is recommended to enhance and strengthen the conclusions presented in the final paragraph of the results and discussion section.
Author Response
Thank you for your valuable time, careful reading of the manuscript and valuable guidance given in review. The content of the manuscript has been greatly improved. Below I am sending responses to the comments contained in the review.
- Enhance the introduction. A short paragraph should be included to complement the information justifying the use of lactic acid bacteria strains, especially Lacticaseibacillus casei and L. plantarum, and how they can influence the wheat flour fermentation processes, conferring the properties and characteristics outlined in the study. Additionally, it is essential to include the chemical changes that enable the development of compounds with potential antioxidant properties in the fermented dough.- We have thoroughly revised the introduction to incorporate the requested enhancements. New information has been included to provide a justification for the selection of LAB strains and their potential influence on wheat flour fermentation processes. Hopefully, these additions strengthen the introduction and provide a clearer context for the study.
- Organize the presentation of the materials and methods section into subsections based on each procedure and activity, rather than presenting it as a single body of text. This organization will improve readability and maintain order. The same observation applies to the presentation of results, which should also be better organized, following the structure outlined in the materials and methods section.- We have restructured both the Materials and Methods section and the Results section to improve readability and maintain better organization. We believe this reorganization enhances the manuscript's overall structure and will make it more accessible to readers and reviewers.
- Figure 4 lacks units corresponding to the Y-axis.- We have revised the figure to include the appropriate units corresponding to the Y-axis.
- Furthermore, the discussion regarding the various factors affecting antioxidant activity should be improved. The phenolic content appears to have a low correlation with the presented antioxidant activity, suggesting that it may depend more on other compounds. In this regard, the authors are advised to thoroughly review the explanations provided, aiming for better alignment with the results of the ABTS and FRAP assays.- We have taken your advice into consideration and have made significant improvements to this section. These revisions provide a more accurate and comprehensive discussion, enhancing the manuscript's overall quality and clarity.
- It is recommended to enhance and strengthen the conclusions presented in the final paragraph of the results and discussion section.- We have revised this part of the manuscript and made significant improvements to provide a more concise summary of our key findings.
Thank you again for your valuable comments.
Reviewer 2 Report
Manuscript ID: foods-2608812
Title
Enhancing Bread's Benefits: Investigating the Influence of Boosted Native Sourdough on FODMAP Modulation and Antioxidant Potential in Wheat Bread
In this study the authors analyzed the impact of two strains of different bacterial species used for sourdough fermentation and the duration of fermentation on the quality of wheat bread, FODMAP content, and antioxidant activity.
Strengths
The experimental design is well done, and the authors have included a suitable array of assays whose results allow for a more comprehensive analysis of the potential of some bacteria to reduce the FODMAP and to develop healthier wheat bread products with enhanced antioxidant activity.
Limitations and suggestions
The novelty of the manuscript is not remarkable high due to the fact that the role of different strains of L. plantarum to improve the quality and shelf life of wheat bread have already been studied. However, the contribution of L. casei strains is less known, and the manuscript could be accepted for publication after some revisions.
The main issue is although some mechanisms or effects might be widespread or frequently observed among most strains of a given Lactic acid bacteria species, others may be strain-specific or strain related ones (i.e., present only in a few strains of a given species). This is the reason for the need to assess several strains to accumulate evidence. In my opinion, the authors should avoid unnecessary generalizations, and highlight the effects observed for the two DSM strains studied. Please, consider to rephrase lines 24, 67-68 and 368-369.
Also, the reference to a previous work using the same strains to study the dynamics of transformations during sourdough fermentation is missing, and should be added and properly discussed.
Minor suggestions:
Lines 91-96: Why authors shift among such different temperatures? (40°C for the initial microbial growth in MRS, kind of restrictive temperature; 28°C for sour soups fermentation and 30°C for sourdoughs fermentation)
Lines 93-94: please, express the inoculum as CFU (colony forming units)
Line 212: the word “control” is misspelled
Lines 214, 254, 288: please, consider to eliminate the word “bacteria” after L. plantarum, it is redundant
Lines 288-290: seems to be true for L. plantarum, not so sure for spontaneous
Line 314-316: Please, consider to add more information of Fructose metabolism in L. casei (a facultative heterofermentative LAB) to clarify its role in mannitol accumulation.
Author Response
Thank you for your valuable time, careful reading of the manuscript and valuable guidance given in review. The content of the manuscript has been greatly improved.
- The main issue is although some mechanisms or effects might be widespread or frequently observed among most strains of a given Lactic acid bacteria species, others may be strain-specific or strain related ones (i.e., present only in a few strains of a given species). This is the reason for the need to assess several strains to accumulate evidence. In my opinion, the authors should avoid unnecessary generalizations, and highlight the effects observed for the two DSM strains studied. Please, consider to rephrase lines 24, 67-68 and 368-369.- Thank you for your valuable insight into the potential strain-specific variations within LAB species. We have carefully rephrased indicated fragments in our manuscript to avoid unnecessary generalizations and to better emphasize the effects observed for the two strains studied.
- Also, the reference to a previous work using the same strains to study the dynamics of transformations during sourdough fermentation is missing, and should be added and properly discussed.- We have now included this reference in the manuscript to connect with the previous study and our current research. We appreciate your suggestion, which has enhanced the completeness of our manuscript.
- Lines 91-96: Why authors shift among such different temperatures? (40°C for the initial microbial growth in MRS, kind of restrictive temperature; 28°C for sour soups fermentation and 30°C for sourdoughs fermentation)- The temperatures selected depend on the protocols used and are selected based on previous research. The phase yields, the fermentation time and the desired amount of CFU and/or the pH of the intermediate were taken into account.
- Lines 93-94: please, express the inoculum as CFU (colony forming units)- corrected.
- Line 212: the word “control” is misspelled- corrected.
- Lines 214, 254, 288: please, consider to eliminate the word “bacteria” after L. plantarum, it is redundant- corrected.
- Lines 288-290: seems to be true for L. plantarum, not so sure for spontaneous- corrected.
- Line 314-316: Please, consider to add more information of Fructose metabolism in L. casei (a facultative heterofermentative LAB) to clarify its role in mannitol accumulation.- We have expanded on this topic.
Thank you again for your valuable comments.
Reviewer 3 Report
The manuscript entitled “Enhancing Bread's Benefits: Investigating the Influence of Boosted Native Sourdough on FODMAP Modulation and Antioxidant Potential in Wheat Bread” provides an important topic related to the practise of improving the quality of wheat bread. However, this article is not well structured and is very scattered. Therefore, I suggest some comments below to improve it:
- In the abstract section, the result of antioxidant activity is not reported. It is advisable to indicate this in order to understand the increase in the value obtained.
- In the Materials and Methods and Results and Discussion sections, subparagraphs are missing. It is advisable to divide these sections into subparagraphs, in order to get a more schematic idea of the experiments carried out.
- In the Materials and Methods section, ABTS and FRAP are mentioned as antioxidant methods for the evaluation of anti-radical activity. It is advisable to correct these definitions, as ABTS is an assay that evaluates anti-radical activity, while FRAP measures the antioxidant activity of compounds capable of reducing the ferric complex.
- It is recommended to add a paragraph about the conclusions.
- The conclusions in the "results and discussions" paragraph are unclear and do not highlight the result obtained and the novelty of this work compared to other previously published papers. It is advisable to highlight the novelty of the work both in the conclusions and introduction sections.
- In Figure 4, the trend of mannitol in the different samples is negligible. It is advisable to indicate this more clearly in the graph.
- In Table 2, indicate in the caption the meaning of the values ​​expressed (L, a, b etc..).
- In Figure 3, indicate the meaning of “d.m.”. This is not reported in the paper.
An English review by a native speaker is recommended.
Author Response
Thank you for your valuable time, careful reading of the manuscript and valuable guidance given in review. The content of the manuscript has been greatly improved.
- In the abstract section, the result of antioxidant activity is not reported. It is advisable to indicate this in order to understand the increase in the value obtained.- Thank you for your suggestion regarding the abstract section. We have now included a brief mention of the antioxidant activity results in the abstract.
- In the Materials and Methods and Results and Discussion sections, subparagraphs are missing. It is advisable to divide these sections into subparagraphs, in order to get a more schematic idea of the experiments carried out.- Thank you for your suggestion to add subparagraphs to the Materials and Methods and Results and Discussion sections for improved organization. We have revised both sections, incorporating subparagraphs to create a more structured and schematic presentation of the experiments and their outcomes.
- In the Materials and Methods section, ABTS and FRAP are mentioned as antioxidant methods for the evaluation of anti-radical activity. It is advisable to correct these definitions, as ABTS is an assay that evaluates anti-radical activity, while FRAP measures the antioxidant activity of compounds capable of reducing the ferric complex.- Thank you for pointing out the need for clarification in the Materials and Methods section regarding the definitions of ABTS and FRAP assays. We have made the necessary corrections.
- It is recommended to add a paragraph about the conclusions.- Thank you for your suggestion to include a paragraph about the conclusions in our manuscript. We have now added a dedicated concluding paragraph to summarize the main findings, highlight the significance of our results, and discuss the potential implications and limitations of our research.
- The conclusions in the "results and discussions" paragraph are unclear and do not highlight the result obtained and the novelty of this work compared to other previously published papers. It is advisable to highlight the novelty of the work both in the conclusions and introduction sections.-. We have revisited both the conclusions and introduction sections to better emphasize the uniqueness and significance of our research in comparison to previously published papers.
- In Figure 4, the trend of mannitol in the different samples is negligible. It is advisable to indicate this more clearly in the graph.- Thank you for your observation regarding Figure 4. To more accurately reflect the negligible trend of mannitol in the different samples, we have made adjustments to the graph.
- In Table 2, indicate in the caption the meaning of the values ​​expressed (L, a, b etc..).- The meaning of colour values indicated in Table 2 are described in point 2.4. in Materials and Methods section.
- In Figure 3, indicate the meaning of “d.m.”. This is not reported in the paper.- Thank you for pointing out the missing explanation for "DM" in Figure 3. We have now added a clear explanation to the figure caption, indicating that "DM." stands for "dry matter".
- An English review by a native speaker is recommended.- The text has been linguistically proofread and the translator's certificate is attached to the system.
Round 2
Reviewer 1 Report
The quality of the manuscript was improved. The suggestions made were included.
Reviewer 3 Report
The manuscript was adjusted correctly according to the reviewer's comments.